# Effect of Bacterial or Fungal Phytase Supplementation on the Performance, Egg Quality, Plasma Biochemical Parameters, and Reproductive Morphology of Laying Hens

**DOI:** 10.3390/ani11020540

**Published:** 2021-02-19

**Authors:** Ahmed A. Saleh, Mohammed Elsawee, Mohamed M. Soliman, Reyad Y. N. Elkon, Mohammed H. Alzawqari, Mustafa Shukry, Abdel-Moneim Eid Abdel-Moneim, Hatem Eltahan

**Affiliations:** 1Department of Poultry Production, Faculty of Agriculture, Kafrelsheikh University, Kafrelsheikh 333516, Egypt; mfayd222@gmail.com (M.E.); reyadnofal@agr.kfs.edu.eg (R.Y.N.E.); m.alzawqari@gmail.com (M.H.A.); 2Clinical Laboratory Sciences Department, Turabah University College, Taif University, Taif 21944, Saudi Arabia; mmsoliman@tu.edu.sa; 3Department of Animal Production, Faculty of Agriculture and Veterinary Medicine, Ibb University, Ibb 70270, Yemen; 4Department of Physiology, Faculty of Veterinary Medicine, Kafrelsheikh University, Kafrelsheikh 33511, Egypt; mostafa.ataa@vet.kfs.edu.eg; 5Biological Application Department, Nuclear Research Center, Egyptian Atomic Energy Authority, Abu-Zaabal 13759, Egypt; aeabdelmoneim@gmail.com; 6Animal Production Research Institute, Agricultural Research Center, Ministry of Agriculture, Dokki 12622, Egypt; hatem_eltahan2002@yahoo.com

**Keywords:** phytase sources, performance, egg quality, blood biochemical parameters, reproductive morphology, layers

## Abstract

**Simple Summary:**

The present study shows that 5000 FTU/kg dietary supplementation with bacterial (*E. coli*) or fungal (*Aspergillus niger* and *Trichodermareesei*) sources of phytase with less available phosphorus is capable of maintaining productive efficiency, reproductive morphology, and egg quality of laying hens. Eggshell consistency was increased while yolk cholesterol was decreased as a result of diets supplemented with bacterial or fungal phytase. All in all, our results clarify that feeding laying hens bacterial and fungal phytase at 5000 FTU/kg can be effective to replace inorganic phosphorus commercially.

**Abstract:**

Catalytic and physicochemical properties of microbial phytase sources may differ, affecting phosphorus (P) release and subsequently the productive and reproductive performance of layers. The current study aimed to evaluate the impact of bacterial and fungal phytase sources on layer productivity, egg production, biochemical blood indices, and reproductive morphology. For this purpose, 360 Bovans brown hens at 42 weeks of age were randomly allocated into 4 experimental groups, each with 15 replicates of 6 hens. The first group (control) was fed a basal diet with 4.6 g/kg available P. In contrast, the second, third, and fourth groups were fed diets treated with 3.2 g/kg available P, supplemented with either 5000 FTU/kg of bacterial *E. coli* (Quantum^TM^ Blue 5G), fungal *Aspergillus niger* (VemoZyme^®^ F 5000 Naturally Thermostable Phytase (NTP)), or fungal *Trichodermareesei* (Yemzim^®^ FZ100). Dietary supplementation of bacterial and fungal phytases did not affect the productive performance or egg quality criteria, except for increased shell weight and thickness (*p* < 0.05). Serum hepatic function biomarkers and lipid profiles were not altered in treated hens, while calcium and P levels were increased (*p* < 0.05) related to the controls. Ovary index and length, and relative weight of oviduct and its segments were not influenced. The contents of cholesterol and malondialdehyde in the yolks from treated birds were lower compared to control hens, while calcium and P content increased (*p* < 0.05). Conclusively, bacterial and fungal phytase sources can compensate for the reduction of available P in layers’ diets and enhance shell and yolk quality without affecting productive performance, and no differences among them were noticed.

## 1. Introduction

Phosphorus (P) is an essential and vital nutrient that is well expressed in the components used in the formulation of feed for poultry production, mainly soybean meal or maize. Nevertheless, two-thirds of the total phosphate content in plant seeds is stored as phytate P and excreted without being digested [1,2]. Phosphorus phytate is poorly absorbed in the gastrointestinal tract of monogastric animals and can additionally negatively affect the digestibility of other nutrients and poultry performance due to its anti-nutritional impact [3,4]. Although sources of inorganic P, such as dicalcium phosphate, can be added to poultry diets, this will increase the amount of released P in the manure. Contamination of water with excessive P leads to its eutrophication and subsequent enrichment of surface water with phytonutrients, which is considered a form of pollution [5]. This problem has motivated scientists to search for suitable ways to decrease the quantity of P excreted in poultry waste, and among these solutions is including phytase in the diet.

In the past few decades, the development of exogenous phytases has been considered one of the most significant findings in poultry nutrition. Exogenous phytases can eliminate the anti-nutritional effect of phytate, allow for better use of P and calcium (Ca), enhance nutrient digestibility, and reduce environmental pollution by P by reducing its excretion in manure [6,7]. Dietary supplementation with exogenous phytases can reduce the required amount of non-phytic P. This elevates the amount of hydrolyzed phytate P available to birds by hydrolyzing the phosphodiester bonds of phytates [8]. Phytases are classified into two subclasses, 3-phytase or 6-phytase, based on the position of the first hydrolyzed phosphate group on the myo-inositol ring [9]. They can also be organized as bacterial or fungal depending on their industrial production process or the microorganisms used as gene donors [10]. Commercially, several types of phytase are available; some are derived from fungi such as *Aspergillus niger* and *A. ficum*, which are generally 3-phytase (EC 3.1.3.8), and others are derived from the bacterium *Escherichia coli* or the fungi *Peniphoralycii* and *Trichodermareesei*, which are 6-phytases (EC 3.1.3.26) [10,11]. Furthermore, the microbial physicochemical and catalytic properties of phytases can differ [12], which can affect P release and subsequently affect the productive and reproductive performance of laying hens. The catalytic properties of phytases differ based on their type: 3-phytases begin the degradation of phytate at position 3, while 6-phytases initiate the release of phosphate radicals in the C6 position of myo-inositol hexaphosphate [10,11]. Moreover, phytase sources may have different characteristics, such as thermal stability, resistance to degradation in the gastrointestinal tract, and pH required for optimal activity [12], which in turn affect the in vivo efficiency of phytase products. In order to assess the efficacy of the new phytases, ongoing studies are required. 

To our knowledge, few studies have investigated the impact of bacterial or fungal phytases on laying hens. Therefore, this experiment aims to evaluate the impact of bacterial or fungal phytase sources on productive efficiency, egg quality, biochemical blood indices, and ovary formation in layers.

## 2. Materials and Methods

### 2.1. Ethical Statement

This study was conducted under the approval of the Ethics Committee of the Local Experimental Animals Treatment Committee and operated following the guidelines of Kafrelsheik University, Egypt (number 4/2016EC).

### 2.2. Birds and Experimental Diets

For the experiment, 360 Bovans brown hens (42 weeks of age, 86% egg production) were caged (105 × 65 × 50 cm^3^ (length × width × height)) in an open-sided building, divided into 4 experimental groups, each with 15 replicates (6 hens/replicate) and under a 16:8 h light:dark lighting regimen with the same environmental management conditions (temperature, moisture, ventilation). Each cage was equipped with stainless steel nipple drinkers providing drinking water ad libitum. Feed in mash form was also offered ad libitum and the ambient temperature ranged between 24 and 27 °C. The first treatment group was fed the basal diet, which contained 4.6 g/kg available P, as control; the second, third, and fourth treatment groups were fed diets that included 3.2 g/kg available P, supplemented with 5000 FTU/kg diet of bacterial phytase (Quantum^TM^ Blue 5G, AB-Vista, Plantation, FL, USA), fungal phytase (VemoZyme^®^ F 5000 NTP, VEMO 99 Ltd., Sofia, Bulgaria), or fungal phytase (Yemzim^®^ FZ100, OrbaBiokimya, Istanbul, Turkey). The first phytase source was derived from *E. coli* (6-phytase; EC 3.1.3.26), while the second and third phytase sources were derived from Aspergillus niger (3-phytase; EC 3.1.3.8) and Trichodermareesei (6-phytase; EC 3.1.3.26), respectively. Each phytase source contained 5000 phytase units (FTU)/kg as described by the manufacturer’s manual, moreover, the three enzymes are thermally stable and the activity recover more than 100% in mash feed 95–99% in pelleted feed; the single unit corresponds to the enzyme quantity needed to liberate 1 μmol of inorganic P from 5.1 mM sodium phytate/minute at 37 °C and pH 5.5 [13]. The composition of the experimental diet is presented in Table 1. Diets were determined to follow the guidelines for Bovans brown layers [14].

### 2.3. Performance

At the beginning (42 weeks old) and the end (54 weeks old) of the experiment, the body weight (BW) of birds was recorded. Eggs were collected daily, weighed individually, and used for all experimental periods to measure mean egg weight. Egg mass was determined by multiplying the production of eggs (% hens/day) by the egg weights. Daily feed intake (FI) was assessed on a cage basis, and the overall feed conversion ratio (FCR) was evaluated as kg of feed per kg of eggs.

### 2.4. Egg Quality

Egg and eggshell quality criteria including egg weight, length, and width, shell thickness, yolk width, thick and thin albumin width, yolk height, egg white height, yolk weight, egg white weight, yolk color, and shell weight were measured at 42 and 54 weeks. From each test, 30 eggs laid between 08:00 and 12:00 were randomly chosen. Eggs were individually weighed and split on the Egg Quality Measure (EQM) plate measuring stand and the albumin height was determined. The strength of the yolk color was measured and reported using the Roche yolk fan color process. Eggshells were cleaned of any adhering albumin to determine shell weight, and the membrane was eliminated; eggshells were then dried at 37 °C for 1 h and their weight was measured as a percentage of the entire egg. An assessment of egg consistency was carried out on individual eggs, equivalent to the estimated egg weight.

### 2.5. Reproductive Morphological Measurements

One bird per replicate was weighed and slaughtered at the end of the experiment. The weight of internal organs and morphology of the reproductive system were evaluated. The weight of liver, gizzard, spleen, and abdominal fat was assessed. The ovary and oviduct were also weighed, and the overall length of the oviduct was measured. The relative weight and length of oviduct portions (including vagina, uterus, isthmus, magnum, and infundibulum) were calculated [15].

### 2.6. Serum Biochemical Parameters

Blood samples were collected and centrifuged for 30 min at 3000 g, and samples for serum constituent measurements were stored at −20 °C. The quantities of total cholesterol, high-density lipoprotein (HDL), albumin, total protein, aspartate aminotransferase (AST), alanine aminotransferase (ALT), Ca, P, and glucose were analyzed spectrophotometrically (Unico UV-2000; SpectraLab Scientific Inc., San marcos, CA, USA) at a wavelength of 545 nm [16] using a commercial kit (Egyptian Company for Biotechnology, Cairo, Egypt). Triglyceride concentration was measured (Wako Chemicals, Richmond, VA, USA) and used as a marker of circulating low-density lipoprotein (LDL), which is an accurate and simple method to estimate the LDL production and reproductive status of domestic fowl [17]. 

### 2.7. Yolk Chemical Analysis

Yolk cholesterol was established in fat extracted from the yolk with a mixture of methanol and chloroform (1:2 vol:vol) [18,19]. Malondialdehyde (MDA) content of the yolk was determined colorimetrically, as described by Saleh et al. [20]. Yolk concentrations of calcium and phosphorus were evaluated by the mineral measurement method [21].

### 2.8. Data Analysis

The collected data were analyzed using the general linear model (GLM) procedure in SAS (v 9.4; SAS Institute, 2016, Gary, CA, USA). The significance of differences of means was tested using Tukey’s test, and all differences were considered significant at *p* < 0.05.

## 3. Results

### 3.1. Performance Measurements

As presented in Table 2, no significant effects were recorded in final BW, FI, FCR, egg weight, egg production, and egg mass of hens fed diets with low available P and fortified with bacterial or fungal sources of phytase compared to the control birds.

### 3.2. Egg Quality

The impact of bacterial and fungal phytases on egg quality characteristics of layers at 42 and 54 weeks of age are shown in Table 3. No significant variations were recorded among egg quality characteristics at 42 weeks. Both bacterial and fungal phytases increased (*p* < 0.05) shell weight and shell thickness compared to the control. Dietary supplementation with Quantum^TM^ Blue 5G reduced (*p* < 0.01) yolk weight and increased (*p* < 0.05) white weight, while VemoZyme^®^ F 5000 NTP and Yemzim^®^ FZ100 did not affect these parameters.

### 3.3. Serum Biochemical Parameters

Results concerning the impact of bacterial and fungal phytases on biochemical blood indices of laying hens are shown in Table 4. Dietary supplements did not significantly alter serum protein fractions, lipid profile, and liver enzymes. Calcium and P levels were increased (*p* < 0.05) in birds supplemented with bacterial and fungal phytases.

### 3.4. Reproductive Morphology Measurements

The data presented in Table 5 show that dietary supplementation with bacterial or fungal phytases did not affect the relative weights of ovary, liver, gizzard, and spleen. The length and relative weight of the oviduct and its segments were also not altered, except for infundibulum weight (%), which was reduced (*p* < 0.05) in groups treated with Quantum^TM^ Blue 5G and Yemzim^®^ FZ100 compared to the control group.

### 3.5. Yolk Chemical Analysis

As shown in Table 6, the MDA content of yolk was numerically reduced in Quantum^TM^ Blue 5G and Yemzim® FZ100 treated groups, while cholesterol content was substantially (*p* < 0.01) reduced in all treated groups. Ca and P concentrations in the yolk were elevated (*p* < 0.05) in Quantum^TM^ Blue 5G and Yemzim® FZ100 compared to the control group.

## 4. Discussion

The beneficial role of bacterial and fungal phytases in alleviating the negative impact of dietary P deficiency in poultry has been well documented [10,22]. This study proposes that phytase can hydrolyze phosphoester bonds of phytates and release nutrients, which improves their utilization. The results of the present study revealed no significant gaps in productivity efficiency between layers fed the control diet and those fed diets with low available P fortified with bacterial and fungal phytases, which appears to support the conclusions of previous studies. Our findings follow those of Shet et al. [23], Tischler et al. [24], and Żyła et al. [25], who described that dietary addition of exogenous phytases at 250, 300, and 500 FTU/kg feed with normal or low levels of available P had no effect on egg number, FI, and FCR. On the contrary, Jalal and Scheideler [26] reported that feeding on a low 0.01% non-phytate P diet supplemented with commercial phytase (250 and 300 FTU/kg) improved the FI, FCR, and egg mass of layers. Moreover, in the present study, we found that no discrepancies between bacterial and fungal phytase sources were noticed. Yan et al. [27], Sands et al. [28], and Stahl et al. [29] showed comparable results in young pigs, broilers, and aged laying hens and explained that these findings were due to improper gastrointestinal pH, elaborating the potential catalytic differences among phytase sources. Additionally, it has been stated that changes in the gastrointestinal maturation of laying hens that occur as the birds get older may affect the potential activity of bacterial and fungal phytases [30,31].

Egg quality characteristics of hens fed diets with low available P and microbial phytases were not different than those fed diets with normal available P. However, eggshell quality criteria (thickness and weight) were improved in treated layers compared to controls. Inconsistency was noticed in the results of egg quality criteria as affected by dietary phytase supplementation. On the one hand, Mohammed et al. [32] reported that supplementation with phytase in layers’ diets had no significant effect on egg quality measurements. Lucky et al. [33], Panda et al. [34], and Cabuk et al. [35] noted no changes in eggshell weight and thickness with various dietary phytase levels. On the other hand, it has been stated that low levels of Ca and available P in diets without phytase supplementation reduced eggshell weight [36,37]. Liu et al. [38] demonstrated that eggshell thickness was improved by dietary supplementation with Ca, P, and phytase. Furthermore, as reported by Żyła et al. [39], Hughes et al. [40], and Jalal and Scheideler [26], dietary phytase was effective in enhancing eggshell quality when laying hens received diets with inadequate non-phytate P. The increased eggshell quality may be attributed to increased nutrient digestibility associated with phytase supplementation and better utilization of P by hens [27].

Reproductive morphology measurements were not influenced by feeding on diets with low available P complemented with bacterial or fungal phytase sources. No previous investigations evaluated the impact of microbial phytases on oviduct and ovarian morphology. The potential role of microbial phytases in maintaining reproductive system morphology in hens fed diets with low available P could be through their ability to reduce endogenous amino acid flow, enhance the availability of dietary energy, and enhance the digestibility of protein, phosphorus, and amino acids [41]. Microbial phytases can also dephosphorylate insoluble phytate salts and liberate minerals such as Zn, Mn, Fe, Cu, and Mg [42,43], which might be involved in regulating the development of the oviduct and ovary in layers [15]. Xie et al. [44] confirmed this assumption and documented that Mn has a vital role in regulating hormones that participate in egg production and ovarian development in broiler breeder hens.

Dietary inclusion of bacterial or fungal phytases increased serum and yolk levels of P compared to the control. These data are in agreement with those of Yan et al. [27], who noticed an elevation in serum Ca and P values of layers fed diets with phytase and found that these levels were positively correlated with nutrient digestibility. Rama-Rao et al. [45] also reported a linear increase in serum P in response to phytase addition. In contrast, serum Ca was not elevated linearly, which might be attributed to the antagonistic impact of serum P and Ca. On the contrary, no significant differences in the P and Ca contents of egg yolk were noticed with phytase addition [46]. Moreover, Viveros et al. [47] and Sebastian et al. [48] documented that phytase supplementation of low non-phytate P diets reduced plasma Ca and increased plasma P levels. The increased serum Ca and P levels can be attributed to the ability of microbial phytases to dephosphorylate phosphoester bonds of phytates and insoluble phytate salts and release P [42,43]. Moreover, it has been reported that feeding on low P diets led to elevate plasma levels of ionized Ca [49]. Generally, numerous factors can affect Ca and P metabolism and deposition in the egg yolk, such as the age and genetic line of the birds and their physiological status.

In the present study, cholesterol concentration in egg yolk was reduced in groups with microbial phytase supplementation. Our results are in line with those of Saleh [50], who reported that the content of total cholesterol in egg yolk was decreased by dietary supplementation with bacterial phytase in low P diets. Furthermore, Żyła et al. [39] also reported that yolk cholesterol was reduced in Hi-sex laying hens fed diets with phytase. The authors suggested that the increased content of unsaturated fatty acids in egg yolk by phytase inclusion might be the cause of the reduction in cholesterol. However, this finding requires more detailed research, mainly because similar reductions in probiotics have been observed [19,51,52,53,54], and probiotics are known to have the ability to produce phytase [55,56].

## 5. Conclusions

The results of the present study demonstrate that 5000 FTU/kg dietary supplementation with bacterial (*E. coli*) or fungal (*Aspergillus niger* and *Trichodermareesei*) sources of phytase is capable of maintaining the productive performance, reproductive morphology, and egg quality of layers with supplementation of 30% less available P. Eggshell quality and yolk cholesterol were also improved in groups fed bacterial or fungal supplemented diets. No obvious differences were noticed in the ability of bacterial and fungal phytase to liberate phytate-bound complexes when supplemented in layers’ diets with 0.32% available phosphorus. Altogether, our findings elucidate that feeding laying hens any studied phytase source at 5000 FTU/kg can be efficient as a substitute for inorganic phosphorus on the commercial scale.

## Figures and Tables

**Table 1 animals-11-00540-t001:** Composition and nutrients of experimental diets.

Ingredient, g/kg	Dietary Phytase, g/kg *			
	Control	Quantum^TM^ Blue 5G	VemoZyme^®^ F 5000 NTP	Yemzim^®^ FZ100
Yellow corn	606	608	608	608
Soybean meal, 46%	203	205	205	205
Corn gluten meal, 62%	62	60	60	60
Soybean oil	5	5	5	5
Dicalcium phosphate	15	9	9	9
DL-methionine, 99%	1	1	1	1
L-lysine, 98%	1.25	1.25	1.25	1.25
Limestone	95	99	99	99
NaCl	2.75	2.75	2.75	2.75
Premix ^1^	3	3	3	3
Sodium bicarbonate	1.25	1.25	1.25	1.25
Potassium carbonate	4.5	4.5	4.5	4.5
Antitoxin ^2^	0.25	0.25	0.25	0.25
Calculated nutrient ^3^				
Crude protein, %	17.49	17.49	17.49	17.49
ME, Kcal/kg diet	2851	2851	2851	2851
Calcium, %	3.26	3.26	3.26	3.26
Total phosphorus, %	0.71	0.57	0.57	0.57
Available phosphorus, %	0.46	0.32	0.32	0.32
Phytate phosphorus, %	0.25	0.25	0.25	0.25
Ether extract, %	4.46	4.46	4.46	4.46
Crude Fiber, %	2.80	2.80	2.80	2.80
Lysine, %	0.88	0.88	0.88	0.88
Methionine, %	0.49	0.49	0.49	0.49
Methionine + cysteine, %	0.771	0.771	0.769	0.770
Threonine, %	0.66	0.66	0.66	0.66

^1^ Premix content: vitamin mineral premix (units per kilogram of feed): vitamin A, 10,000 IU; vitamin D3, 3,500 IU; vitamin E, 35 IU; menadione, 1.5 mg; riboflavin, 5 mg; pantothenic acid, 8 mg; vitamin B12, 0.012 mg; pyridoxine, 1.5 mg; thiamine, 1.5 mg; folic acid, 0.5 mg; niacin, 30 mg; biotin, 0.06 mg; iodine, 0.8 mg; copper, 10 mg; iron, 80 mg; selenium, 0.3 mg; manganese, 80 mg; zinc, 80 mg. ^2^ Calculated according to guidelines for brown Bovans laying hens [14]. 2Antitoxin; Hydrated sodium calcium aluminosilicates (HSCAS); ^3^ Calculated according to ingredients composition provided by National Research Council (1994). * Control diet was not supplemented with phytase; other diets were supplemented with phytase sources at 100 mg phytase/kg diet. The enzymes activities are 100% recovered according to feed analyzed by the three enzymes companies’ methods of analysis the phytase activity. NTP: Naturally Thermostable Phytase.

**Table 2 animals-11-00540-t002:** Effect of bacterial or fungal sources of phytase on performance in layers.

Item	Dietary Phytase, FTU/kg *				SEM	*p*-Value
	Control	Quantum^TM^ Blue 5G	VemoZyme^®^ F 5000 NTP	Yemzim^®^ FZ100		
Initial body weight, g	1799	1800	1812	1814	11.95	0.7083
Final body weight, g	1881	1880	1893	1894	12.14	0.7791
Body weight gain, g/12 weeks	82.7	79.7	81.0	79.3	4.30	0.9561
Feed intake, g/day	106.8	108.5	104.7	105.0	1.05	0.4270
Egg weight, g	61.64	62.60	62.32	62.35	0.46	0.4958
Egg production, %	89.60	89.46	87.94	88.13	1.29	0.7163
Egg mass, g of hen^−1^ day^−1^	55.22	55.97	54.82	54.91	0.81	0.7460
Feed conversion ratio, g feed/g egg	1.94	1.95	1.91	1.92	0.02	0.6517

Values presented are means and standard errors of 60 per treatment. * Control diet was not supplemented with phytase; other diets were supplemented with phytase sources at 5000 FTU/kg. SEM, standard error of means. NTP: Naturally Thermostable Phytase.

**Table 3 animals-11-00540-t003:** Effect of bacterial or fungal phytase sources on egg quality in layers.

Item	Dietary Phytase, FTU/kg *				SEM	*p*-Value
	Control	Quantum^TM^ Blue 5G	VemoZyme^®^ F 5000 NTP	Yemzim^®^ FZ100		
At 42 weeks						
Egg weight, g	60.40	60.40	60.50	60.00	0.14	0.0816
Egg length, mm	55.6	54.6	55.2	55.84	0.04	0.2648
Egg width, mm	44.7	43.8	44.2	43.4	0.06	0.4348
Shell thickness, mm	574.64	569.00	572.00	571.00	8.72	0.2639
Yolk width, mm	4.01	3.98	4.14	4.04	0.07	0.4125
Thick albumin width, mm	3.01	3.23	4.12	3.68	0.31	0.0707
Thin albumin width, mm	2.72	2.74	3.25	3.52	0.29	0.1509
Yolk height, mm	18.24 ^a^	17.82 ^ab^	17.04 ^ab^	16.50 ^abc^	0.29	0.0005
Albumin height, mm	7.31 ^a^	7.50 ^a^	6.46 ^ab^	5.93 ^ab^	0.43	0.0442
Yolk weight, g	15.00	14.90	15.60	16.00	0.52	0.4086
White weight, g	38.00	37.90	37.70	36.40	0.60	0.2162
Shell weight, g	7.00	7.20	6.80	7.20	0.25	0.6192
Yolk color, Roche fan	6.20	5.80	6.40	6.20	0.22	0.2812
At 54 weeks						
Egg weight, g	64.70	64.80	64.60	65.30	0.27	0.2803
Egg length, mm	5.73	5.61	5.64	5.63	0.07	0.1300
Egg width, mm	4.46	4.49	4.47	4.51	0.02	0.0958
Shell thickness, mm	527.00 ^c^	562.96 ^a^	551.97 ^b^	551.63 ^b^	9.28	0.0015
Yolk width, mm	3.99	3.88	4.02	3.95	0.05	0.3038
Thick albumin width, mm	1.92	2.32	1.78	2.08	0.29	0.6224
Thin albumin width, mm	1.57	1.12	1.74	1.25	0.31	0.4724
Yolk height, mm	18.20	18.01	17.63	18.29	0.20	0.1048
Albumin height, mm	7.90	8.11	7.60	8.39	0.40	0.5688
Yolk weight, g	16.20 ^a^	14.30 ^b^	15.60 ^ab^	15.10 ^ab^	0.37	0.0077
White weight, g	40.70 ^b^	42.80 ^a^	41.70 ^ab^	42.60 ^ab^	0.51	0.0236
Shell weight, g	6.40 ^b^	7.97^a^	7.20 ^a^	7.89a	0.22	0.04092
Yolk color, Roche fan	6.60	6.90	6.50	6.60	0.15	0.2839

Values presented are means and their standard errors of 30 eggs per treatment. ^a,b,c^ Mean values followed by different letters in the same row are significantly different (*p* < 0.05). * Control diet was not supplemented with phytase; other diets were supplemented with phytase sources at 5000 FTU/kg. SEM, standard error of means. NTP: Naturally Thermostable Phytase.

**Table 4 animals-11-00540-t004:** Effect of bacterial or fungal phytase sources on serum biochemical parameters of layers at 54 weeks of age.

Item	Dietary Phytase, FTU/kg *				SEM	*p*-Value
	Control	Quantum^TM^ Blue 5G	VemoZyme^®^ F 5000 NTP	Yemzim^®^ FZ100		
Total protein, mg/dL	5.23	5.55	5.98	6.35	0.35	0.1640
Albumin, mg/dL	2.05	2.23	2.35	2.43	0.14	0.3045
Glucose, mg/dL	148.00	149.75	142.50	135.25	9.11	0.6806
ALT, U/I	3.75	3.60	2.93	3.63	0.40	0.4754
AST, U/I	205.00	247.75	227.25	226.50	21.63	0.5973
Total cholesterol, mg/dL	136.50	138.50	122.50	125.25	9.09	0.5317
Triglyceride, mg/dL	65.72	56.93	57.68	51.50	5.93	0.9100
HDL, mg/dL	31.50	29.00	26.25	29.25	2.72	0.6117
LDL, mg/dL	69.00	76.50	81.75	86.00	8.62	0.5581
Calcium, mg/dL	13.10 ^b^	15.98 ^a^	14.65 ^ab^	15.05 ^a^	1.00	0.0472
Phosphorus, mg/dL	5.08 ^b^	6.78 ^a^	6.48 ^a^	6.35 ^a^	0.32	0.0220

Values presented are means and standard errors of 15 samples per treatment. ^a,b^ Mean values followed by different letters in the same row are significantly different (*p* < 0.05). * Control diet was not supplemented with phytase; other diets were supplemented with phytase sources at 5000 FTU/kg. AST, aspartate aminotransferase; ALT, alanine aminotransferase; HDL, high-density lipoprotein; LDL, low-density lipoprotein; SEM, standard error of means. NTP: Naturally Thermostable Phytase.

**Table 5 animals-11-00540-t005:** Effect of bacterial or fungal phytase sources on internal organ weight and reproductive morphology of layers at 54 weeks of age.

Item	Dietary Phytase, FTU/kg *				SEM	*p*-Value
	Control	Quantum^TM^ Blue 5G	VemoZyme^®^ F 5000 NTP	Yemzim^®^ FZ100		
Live body weight, g	1834.40	1759.80	1802.80	1820.40	19.96	0.0854
Liver weight, g/100 g BW	1.830	1.865	1.783	1.845	0.097	0.9204
Gizzard weight, g/100 g BW	1.079	1.071	1.067	1.080	0.048	0.4604
Spleen weight, g/100 g BW	0.063	0.063	0.067	0.064	0.0011	0.2254
Fat weight, g/100 g BW	3.595	3.602	3.585	3.579	0.174	0.8817
Ovary weight, %	37.60	40.00	41.60	47.60	2.74	0.1043
Oviduct weight, %	69.20	68.60	72.00	62.40	3.95	0.3972
Infundibulum weight, %	2.60 ^a^	1.60 ^b^	2.00 ^ab^	1.00 ^b^	0.36	0.0405
Magnum weight, %	32.00	29.40	30.40	27.60	2.41	0.6328
Isthmus weight, %	6.40	7.00	7.20	4.60	0.86	0.1717
Uterus weight, %	23.80	23.40	25.60	23.80	2.02	0.8685
Vagina weight, %	4.20	6.60	5.40	5.60	0.95	0.3889
Oviduct length, mm	70.60	75.20	73.00	69.20	2.84	0.4756
Infundibulum length, mm	10.00	9.00	10.30	8.10	1.06	0.4666
Magnum length, mm	36.00	40.40	37.00	39.30	2.24	0.5511
Isthmus length, mm	11.90 ^ab^	12.30 ^a^	13.20 ^a^	8.40 ^b^	0.91	0.0095
Uterus length, mm	7.00	7.10	7.80	7.00	0.49	0.6170
Vagina length, mm	4.30	5.40	4.60	4.90	0.49	0.4594

Values presented are means and their standard errors of 30 eggs per treatment. ^a,b^ Mean values followed by different letters in the same row are significantly different (*p* < 0.05). * Control diet was not supplemented with phytase; other diets were supplemented with phytase sources at 5000 FTU/kg. SEM, standard error of means; BW, body weight. NTP: Naturally Thermostable Phytase.

**Table 6 animals-11-00540-t006:** Effect of bacterial or fungal phytase sources on calcium, phosphorus, cholesterol, and malondialdehyde (MDA) contents in egg yolk at 54 weeks of age.

Item	Dietary Phytase, FTU/kg *				SEM	*p*-Values
	Control	Quantum^TM^ Blue 5G	VemoZyme^®^ F 5000 NTP	Yemzim^®^ FZ100		
MDA, nmol/g	0.229	0.188	0.202	0.179	0.013	0.069
Cholesterol, g/g	14.06 ^a^	10.96 ^c^	12.38 ^b^	10.77 ^c^	0.35	0.003
Calcium, mg/100 g	113.7 ^b^	132.2 ^a^	121.3 ^ab^	131.7 ^a^	3.5	0.042
Phosphorus, mg/100 g	267.7 ^b^	301.7 ^a^	279.3 ^ab^	304.6 ^a^	6.8	0.008

Values presented are means and their standard errors of 30 eggs per treatment. ^a,b,c^ Mean values followed by different letters in the same row are significantly different (*p* < 0.05). * Control diet was not supplemented with phytase; other diets were supplemented with phytase sources at 5000 FTU/kg. SEM, standard error of means. NTP: Naturally Thermostable Phytase.

## Data Availability

Data supporting this study’s findings are available by fair request from the corresponding author.

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
