# Peer review of "Effect of Bacterial or Fungal Phytase Supplementation on the Performance, Egg Quality, Plasma Biochemical Parameters, and Reproductive Morphology of Laying Hens"

_animals, 2021, doi:10.3390/ani11020540_

Round 1

Reviewer 1 Report

The authors have addressed all my concens

Author Response

Comments and Suggestions for Authors

Author's Reply to the Review Report (Reviewer 1)

The authors have addressed all my concens

Dear Reviewer thank you so much for your spend time to review the manuscript and we believe that your comments will increase the scientific value of our paper.

Reviewer 2 Report

The manuscript «Effect of bacterial or fungal phytase supplementation on the performance, egg quality, plasma biochemical parameters, and reproductive morphology of laying hens … ) has improved significantly in the new version. Thanks for that. But there are still some aspects that must be changed or considered in more detail.

L 105 ff:         The 3 different phytase products are considered to contain exactly 5’000 FTU/kg diet. Phytases are not pure substances like salt or Dicalcium phosphate. They are extracts from microorganisms. They have usually a higher enzyme activity than exactly 5’000 FTU per kg diet. Furthermore, they may have more or less relevant side effects. This is not considered here. Only in the discussion the authors go into this aspect. Also feed sources can have phytase activities. Therefore, phytase activity of the control diet can also be relevant. To be discussed.

Tab. 1:            What is Antitoxin?
2Calculated nutrient content; add comments in the appendix.

What do you mean with “fiber”?

Add here al least analyzed phytase activity values also for the control diet (you have certainly some feed samples left.

Tab. 2:            Use less digits after the decimal point. You cannot measure the weight of a hen with an accuracy of 10 mg! This holds true for many other parameters (feed intake, . . .)

  1. 197 ff: What do you mean with” despite differences in the physicochemical and catalytic properties of microbial phytases, no discrepancies between bacterial and fungal phytase sources were noticed”? You did not go into this aspect in the introduction or material and methods.
  2. 203 ff: What do you mean here?
  3. 219 ff: Consider here also the better utilization of other nutrients (e.g. Ca). See later line 227 ff and 242 ff!

Author Response

Dear Prof. Editor-in-Chief, Animals

cc,

Prof.  Neaqol Li, Assigned Editor

Regarding to the manuscript entitled "Effect of bacterial or fungal phytase sources  supplementation on the performance, egg quality, plasma biochemical parameters, and reproductive morphology in laying hens" 

Thank you in advance for your time and effort on reviewing our work.

A list of modifications according to the suggestions and comments of the reviewers is attached below. We are fully appreciated the valuable suggestions of the reviewers. Moreover, we are proud that our study has good discussion by the reviewers.

Sincerely Yours,

Ahmed Ali Mahmoud Saleh, PhD

Professor
Department of Poultry Production

Faculty of Agriculture,

Kafrelsheikh University, Egypt.

List of modifications according to the suggestions and comments of reviewers:

(Revisions related to reviewers’ comments are shown in red in the revised manuscript)

The authors appreciate the comments from the reviewers. The manuscript has been revised in accordance with their requests. We do our best to take all comments in consideration, incorporating them into the revised manuscript as indicated in our responses to the reviewers.

Comments and Suggestions for Authors

Author's Reply to the Review Report (Reviewer 1)

The authors have addressed all my concens

Dear Reviewer thank you so much for your spend time to review the manuscript and we believe that your comments will increase the scientific value of our paper.

Author's Reply to the Review Report (Reviewer 2)

Comments and Suggestions for Authors

The manuscript «Effect of bacterial or fungal phytase supplementation on the performance, egg quality, plasma biochemical parameters, and reproductive morphology of laying hens … ) has improved significantly in the new version. Thanks for that. But there are still some aspects that must be changed or considered in more detail.

L 105 ff:         The 3 different phytase products are considered to contain exactly 5’000 FTU/kg diet. Phytases are not pure substances like salt or Dicalcium phosphate. They are extracts from microorganisms. They have usually a higher enzyme activity than exactly 5’000 FTU per kg diet. Furthermore, they may have more or less relevant side effects. This is not considered here. Only in the discussion the authors go into this aspect. Also feed sources can have phytase activities. Therefore, phytase activity of the control diet can also be relevant. To be discussed.

The response;

Thank  you so much for your comments .We agree with your comments that the phytase are not substances but byproduct produced from the fermentation of the microorganisms and the enzymes activity are 5000 phytase units (FTU)/kg according the manufacturer’s manuals. And from the manufacturer’s manuals of the three compaies which produced the enzymes they informed that enzymes are thermally stable and the activity recovers more than 100% in mash feed 95-99% in pelleted feed. We added this L 110-112.

Tab. 1:            What is Antitoxin?

The response;

 The Antitoxin  is Hydrated sodium calcium aluminosilicates (HSCAS). We added in table.1

2Calculated nutrient content; add comments in the appendix.

 We added it in table.1. 3Calculated according to ingredients composition provided by National Research Council (1994).

What do you mean with “fiber”?

 We added it in table.1. Crude Fiber, %

Add here al least analyzed phytase activity values also for the control diet (you have certainly some feed samples left.

 We already analysis the activity recover of the three phytase by analysis the activity after mixed the diets before starting the experiment and found that the enzymes activities are 100% recovered according to feed analyzed by the three enzymes companies’ methods of analysis the phytase activity. Added table.1

Tab. 2:            Use less digits after the decimal point. You cannot measure the weight of a hen with an accuracy of 10 mg! This holds true for many other parameters (feed intake, . . .)

We corrected them in table.2

  1. 197 ff: What do you mean with” despite differences in the physicochemical and catalytic properties of microbial phytases, no discrepancies between bacterial and fungal phytase sources were noticed”? You did not go into this aspect in the introduction or material and methods.

We corrected it.

  1. 203 ff: What do you mean here?

According to the Marounek et al. 2008, they concluded that A relatively high total phytase activity was found in the caeca of both young (347 μmol h−1) and old (632 μmol h−1) hens compared to that in other digestive segments, with little activity in the stomach (47 and 102 μmol h−1, respectively) and intermediate in the small intestine (226 and 264 μmol h−1, respectively).

Marounek, M.; Skřivan, M.; Dlouhá, G.; Břeňová, N. Availability of phytate phosphorus and endogenous phytase activity in the digestive tract of laying hens 20 and 47 weeks old. Anim. Feed Sci. Technol.2008, 146, 353-359.

  1. 219 ff: Consider here also the better utilization of other nutrients (e.g. Ca). See later line 227 ff and 242 ff!

We corrected it.

This manuscript is a resubmission of an earlier submission. The following is a list of the peer review reports and author responses from that submission.

Round 1

Reviewer 1 Report

The line numbers in the manuscript is not clear and I am unable to give specific comments.

The study reported in this manuscript is scientifically sound and appears to have been conducted with due care. The research topic falls within the scope of the Journal and a good addition to existing literature on phytase use in layer diets.

A major issue is that the manuscript is not well written. It suffers from excessive number of minor errors in language and presentation style.  It is not possible for this reviewer to provide a list of these issues – too many errors. It is recognised that English is not the first language of the authors, but the authors are responsible (or to seek the services of English editorial advice) to improve the clarity that will meet the standards required for publication in an international journal.

P 1, L2             ..with % less available P ?

P1, L4              cholesterol … increased? – not correct

P3, Para 1        Location (city) of the manufacturer need be included. Unit must be in U/kg diet, not g/ton – change throughout the text.  Were all phytases added in the same activity?. Why NRC(1994) guidelines used? – 25 years old and for older layer strains with low egg numbers. You must use current breeder guidelines. The differences between phytases in terms of physiochemical and catalytic properties are mentioned several times in the text – but no description provided. These need be clearly stated. Indicate if mash diets were used

Table 1:  Column 1: indicate M+C and Thr levels

Table 2: last column: change ‘p-tukey’ to ‘probability’ – here and elsewhere

Table 3: Column 1:  several terms unclear – revise: At the begging; long axis; accident diameter; white diameter 1; white diameter 2 etc. Clearly define the terms used – confusing – see last paragraph, page 3

Table 5: Column 1: Live wt redundant; absolute weights of liver, gizzard, spleen, fat are meaningless – relate to BW. Other % weights – are they % BW

Weak discussion – without line numbers difficult to give specific comments. Just comparing with published data is not adequate. Be critical and focus on main findings.

Not sure whether referencing style is correct: e.g. Shet, et al. {23}

Reviewer 2 Report

This manuscript describes an experiment with laying hens in which 3 commercial phytase products are compared with a control diet in which the Ca and P levels were increased. The authors probably carried out the experiments with care. But here is the fundamental lack of experimental design. There is no negative control treatment with the reduced mineral content and no phytase supplementation. This deficiency does not allow an interpretation of an independent enzyme effect on an experimental parameter. The only correct statement is that no differences were found between the 3 enzyme products.

Furthermore

The 3 enzyme products are not scientifically described at all
Materials and methods: Missing details in many aspects: bird health, housing, climate,. . .

Analyzed enzyme activity measurements in experimental diets are lacking

Form of experimental diets (mash or pellets, heat treatment)
Not all dimensions correct